# Reduced Low–Pressure Membrane Fouling by Inline Coagulation Pretreatment for a Colored River Water

**DOI:** 10.3390/membranes12111028

**Published:** 2022-10-22

**Authors:** Joseph D. Ladouceur, Roberto M. Narbaitz

**Affiliations:** Department of Civil Engineering, University of Ottawa, Ottawa, ON K1N 6N5, Canada

**Keywords:** membrane fouling, inline coagulation, ultrafiltration, natural organic matter, dissolved air flotation, colored water, multi–channel membrane

## Abstract

Drinking water treatment (DWT) using low–pressure membranes (LPM) has become increasingly popular due to their many reported advantages compared to conventional technologies. Productivity decline due to fouling has prevented LPMs from becoming the technology of choice in DWT, however, coagulation pretreatment either with or without particle separation mitigates fouling phenomena. The effectiveness of coagulation/flocculation/sedimentation (CF–S), coagulation/flocculation/dissolved air flotation (CF–DAF), and inline coagulation (CF–IN) as technologies for pretreatment of feed water has rarely been investigated using the same water source. In this study, CF–S, CF–DAF, and CF–IN are directly compared as pretreatment of a tubular multi–channeled ultrafiltration (UF) membrane using the same highly colored river water. Three–day long filtration tests were performed using an automated bench–scale filtration apparatus with an inside–out configuration. Although CF–DAF had the greatest removal of dissolved organic matter (DOM) and hydrophobic organics, CF–S pretreatment resulted in a similar level of total fouling. Compared to CF–DAF and CF–S, CF–IN pretreatment resulted in lower fouling. The hydraulic and chemical reversibility of CF–IN fouling was seen to be strongly influenced by the feed water zeta potential, suggesting the importance of floc electrostatic and morphological characteristics on inline coagulation performance.

## 1. Introduction

Low–pressure membranes (LPMs) have become increasingly popular in drinking water treatment (DWT) due to their effectiveness in particle and bacterial separation [1,2,3,4,5]. Since the late 1990s, technology advancement, increasingly stringent treated water regulations, increased water demand and scarcity, and degrading source water quality has helped to drive the growth of LPM technologies for DWT [6,7,8]. This growth has been aided by the fact that LPMs are known to be less susceptible to raw water quality fluctuations when compared to deep bed filters and produce superior quality effluent [6,9]. Despite their numerous advantages, the more widespread implementation of LPMs in DWT remains a challenge due to membrane fouling, which results in a progressive worsening of membrane productivity, higher energy costs, and a potential reduction in membrane lifespan [10,11,12]. Particulate matter, dissolved organic and inorganic substances, and microorganisms (i.e., biofouling) have all been reported to cause LPM productivity decline [13,14]. Although synergistic fouling by particulates and natural organic matter (NOM) has been reported in the literature [15,16], NOM has been identified by many researchers as the principle foulant of LPMs [17,18,19]. Accordingly, there has been a concerted effort to develop membranes/membrane materials as well as best practices for LPM operation which help to alleviate fouling by NOM. Concerning LPM best practices, the most common approaches followed have been: (a) adjustment of operating conditions (such as reduction of membrane flux or alteration/optimization of membrane cleaning strategies) [10,20,21,22,23,24]; and (b) implementation of feed water pretreatment processes [25,26]. This article will consider the latter approach.

The selection of a given pretreatment strategy will depend on cost, considerations of integrated design, and most importantly on the raw water quality [27,28]. The selection is further complicated when LPM operating conditions and material/configuration characteristics are considered [29]. For surface waters low in particulate and organic matter, LPM facilities can often efficiently operate without the use of any pretreatment in a scheme known as direct filtration [30,31,32]. Higher particulate and/or organic concentrations, typical of many highly colored river waters, are problematic for LPM systems. For such waters, pretreatment is often necessary to maintain stable and economic operation as well as to achieve enhanced removals of contaminants (i.e., disinfection by–product precursors) within the framework of an integrated design [33,34]. Amongst numerous possible alternatives, pre-coagulation is the most commonly employed pretreatment approach for controlling LPM membrane fouling due to its relative low cost, its reported effectiveness, and its familiarity to plant operators [2,25,26,35,36]. Pre-coagulation of LPM feeds has been performed both with and without particle separation processes [26,37]. For concentrations of particles and/or organics resulting in a moderate fouling load on the membrane surface, coagulation/flocculation (CF–IN) without particle separation (i.e., inline coagulation) can be employed before the LPM [38,39]. If elevated NOM or highly variable turbidity levels exist in the source water, then coagulation/flocculation/sedimentation (CF–S) is often considered for feed water pretreatment [30,40]. For algal impacted waters, dissolved air flotation (DAF) may be selected as the pretreatment separation process as it is more adept for the removal of low–density molecules (like algae) than sedimentation and can accomplish the removal in a smaller footprint [41,42]. 

Feed water pretreatment by CF–IN has garnered increased attention over the past decade [43,44,45], likely due to the cost savings its offers water purveyors through its reduction in facility footprint [46,47]. This is achieved through elimination of the flocculation and sedimentation basins, which are not required since the flocs are delivered to the membrane surface and their size needs only to be larger than the membrane pore, not settleable [48,49,50]. Compared to direct filtration, several authors have found that CF–IN pretreatment can mitigate irreversible membrane fouling [51,52]. Some studies have even reported improved performance at sub-optimal coagulant dosages as low as 0.2 mg L^−1^ as alum [53,54,55]. The relative performance of CF–IN compared to other pretreatment approaches, however, has not been extensively investigated. A limited number of studies comparing CF–DAF and CF–IN [56] or CF–S and CF–IN [7,57,58] are available in the literature, but present conflicting results. Amjad et al. [7], who compared CF–IN and CF–S, and Braghetta et al. [56], who compared CF–DAF and CF–IN, both found that the CF–IN pre-treated water had a higher fouling potential compared to waters whose pretreatment involved a particle separation step. In contrast, Yu et al. [58] reported that CF–S pretreatment resulted in the development of a dense cake layer of low porosity which gave rise to a rapid increase in transmembrane pressure (TMP) compared to CF–IN pretreatment. 

Although there have been number of studies which investigate CF–S [37,38] and CF–DAF [59,60] pretreatments individually, there remains a very limited number of research studies that directly compare their performance for LPM fouling mitigation. The earliest studies comparing CF–DAF and CF–S, which considered pressure–driven inside–out polyethersulfone (PES) hollow fiber membranes and a colored river water, reported that CF–DAF resulted in somewhat better organics removal (both in terms of bulk dissolved organic carbon (DOC) and specific ultraviolet absorbance (SUVA)) and reduced membrane fouling compared to CF–S [61,62]. In our recent work using the same highly colored river water and a high permeability pressure–driven outside–in polyvinylidene fluoride (PVDF) hollow fiber membrane, it was shown that, although CF–DAF pretreatment yielded slightly better removal of organics, both pretreatment schemes (i.e., CF–S and CF–DAF) resulted in very similar levels of fouling [63]. The study also showed that CF–IN pretreatment led to greater fouling than CF–S and CF–DAF pretreatment, likely due to the higher foulant loads imparted on the membrane surface [63].

Based on the above findings, it is of interest to ascertain whether the differences in LPM fouling behavior following pretreatment of highly colored waters were a result of the distinct membrane material/configuration used in each study. Accordingly, the objectives of this study are to: (a) directly compare the fouling of a different PES UF membrane following CF–S, CF–DAF, and CF–IN pretreatment; (b) investigate the suitability of sub-optimal (based on jar testing) coagulant dosages on the performance of the integrated CF–IN–UF membrane system; and (c) assess the impact of seasonality on CF–IN pretreatment performance.

## 2. Materials and Methods

### 2.1. Raw Water

Ottawa River Water (ORW) was selected for the challenge water in order to minimize the effects of water quality differences when comparing to the results of our previous study [63]. ORW is typical of northern waters in the Canadian context but also in many regions throughout the globe [64,65,66,67]. Furthermore, ORW is also representative of waters impacted by climate change [68,69], which are characterized by elevated NOM and color levels due to increased flood and drought cycles as well as changing precipitation patterns [68,70]. Raw ORW samples were collected at the intake of the Britannia water treatment plant (WTP) (Ottawa, Canada). ORW samples were collected during both the winter (ORW_w_) and summer (ORW_s_) season on days where the water quality was representative of average conditions (i.e., not following large rainfall or runoff events). The water samples were transferred to a walk–in refrigerator immediately following collection and were stored in the dark at 4 °C to minimize biological degradation. Prior to the filtration experiments, the water samples were placed in the laboratory for 24 h and allowed to reach room temperature (~20 °C) [71]. Therefore, any observed difference in performance during the seasonal assessment is a result of water quality differences and not temperature. The ORW_w_ sample was used to evaluate membrane fouling for raw ORW, CF–S, CF–DAF and CF–IN pretreated waters while an evaluation of seasonality effects using ORW_s_ was limited to CF–IN pretreatment. 

### 2.2. Filtration Apparatus and Membrane Module

An automated bench–scale filtration apparatus (Figure 1) was used to investigate membrane fouling following pretreatment by different methods. The filtration apparatus possessed two subsystems (filtration and backwash), each of which could be independently programmed using a LabView software (National Instruments, Austin, TX, USA).

Pressure–driven inside–out, dead–end membrane modules (Figure 2a) were fabricated using tubular multi–channeled (TMC) membrane fibers. The modules housed a single hydrophilized PES (h–PES) fiber (Multibore, Inge GmbH, Greifenberg, Germany) and operated with the same inside–out flow pattern as used by the manufacturer in its full–scale modules. The tubular fibers have an outside diameter of approximately 4 mm and consist of 7 concentrically positioned capillaries (each capillary with a diameter of 0.9 mm) with a 300 µm porous foam–like support layer between them (Figure 2b). The pure water permeability of a new membrane fiber is reported to be 700–1000 L m^−2^ h^−1^ bar^−1^ and has a nominal pore size of 0.02 µm [72]. 

Modules were constructed in–house using a single TMC fiber cut to a total length of 20 cm yielding a surface area of approximately 35 cm ^2^ (Figure 2a). The fibers were potted within the 30 cm long module using a high strength epoxy resin. The potting sealed the annular cavity space between the membrane fiber and the module casing at the module entrance (end B in Figure 2a) as well as the fiber capillaries at the module exit (end A in Figure 2a). The highly porous foam–like support structure (hatched region in Figure 2b) located between the fiber capillaries at the module entrance also needed to be sealed to force the challenge water to enter the capillaries (as opposed to the foam structure directly). This was accomplished by placing a very fine layer of epoxy resin on the face of the fiber (hatched area in Figure 2b) at the module entrance with the help of a 21–gauge needle attached to a compressed air line. The needle was inserted into the capillary of the membrane fiber at end A (Figure 2a), with the resulting air flow ensuring that epoxy did not seal the capillary entrance on the membrane face. The potting process forced the challenge water to enter the fiber capillaries at end B (Figure 2a), cross the separation skin and pass through the foam–like support structure, prior to exiting the module at end A (Figure 2a) for collection and analysis. During the hydraulic backwash process, the potting procedure forced the backwash water, which entered at end C (Figure 2a), to pass through the foam–like support structure and separation skin in the reverse direction, prior to exiting the module at end B (Figure 2a). Stereoscopic microscope images (Stemi 305 Stero Microscope, Carl Zeiss, Oberkochen, Germany) verifying the effectiveness of the potting process are shown in Appendix A. Each run was conducted using a newly potted module.

### 2.3. Pretreatment Methods

CF–S pretreated water was collected from the Britannia WTP’s pilot facility, which draws water from the outlet of the sedimentation basin. To achieve optimal removal of organics at the time the winter CF–S sample was collected, the Britannia WTP dosed 34 mg L^−1^ alum (3.09 mg L^−1^ as Al), 8.50 mg L^−1^ sulfuric acid (H_2_SO_4_), and 1.25 mg L^−1^ activated silicate (SiO_2_) to reduce the raw water pH to around 6 ahead of hydraulic flocculation and inclined plate clarifiers. CF–DAF pretreated ORW was produced using a large volume bench–scale dissolved air flotation system (LB–DAF) [73] as local COVID-19 pandemic guidelines prevented access to the nearby full–scale Aylmer DAF facility. To best simulate the optimized conditions at the Aylmer WTP, raw ORW was conditioned using the same chemicals and dosages used at the full–scale facility (i.e., 37 mg L^−1^ alum (Kemira ALS) and 0.1 mg L^−1^ cationic polyacrylamide polymer (Superfloc C–492PWG, Kemira)). All flotation parameters were as recommended by the LB–DAF system developers [73].

CF–IN pretreatment was simulated at the lab–scale using a batch–wise approach [49,53,74]. Aliquots of raw ORW were placed in a 20 L baffled mixing tank and rapidly mixed at G = 296 s^−1^ for 2 min following the addition of sulfuric acid and alum. The pH was adjusted to a value of 6 to match that of the CF–S pretreated water and because it has previously been reported that optimal removal of organics could be achieved near this pH for ORW [75]. Two different coagulant dosages were used: (i) 34 mg L^−1^ as alum, which was the optimal coagulant dose for organics removal based on jar tests and was also the dose used at the Britannia WTP; and (ii) 27 mg L^−1^ as alum (a 20% reduced dose). The latter dose was selected because Konieczny et al. [76] found that for coagulation with alum, the lowest permeability decline was achieved at a 20% reduced coagulant dose compared to the optimal determined from jar testing. Furthermore, Choi and Dempsey [77] found that the acidic under-dosed coagulation condition resulted in the lowest fouling resistance during filtration. It should be noted that the same coagulant doses were used during the winter and summer seasons for CF–IN pretreatment for consistent analysis. Following the rapid mix period, the mixer speed was reduced to provide a constant mean velocity gradient of 200 s^−1^. Like Peleato et al. [53] and Tang et al. [74], this study provided continuous intense mixing to the coagulated feed water in order to prevent fluctuating feed water quality due to the settling of flocs.

### 2.4. Multi–Day Filtration Test Protocol

Filtration conditions used in this study are similar to those used in our previous study [63] and are briefly described below. A three–day duration was selected for all experiments as longer duration filtration tests are reported to be more representative of full–scale behavior [61,78]. Filtration tests for raw ORW, as well as CF–DAF and CF–S pretreated waters, were conducted in replicate using winter water samples, while those evaluating CF–IN pretreatment during both seasons were limited to single runs. For all experiments the filtration flux was fixed at 80 L m^−2^ h^−1^, which was chosen so that the membrane would operate in the subcritical range based on preliminary critical flux tests (Appendix A) conducted using the flux stepping method proposed by Le Clech et al. [79]. Hydraulic backwashing was performed using RO water after every 30 min filtration cycle. Each backwash cycle continued for a duration of 2 min at a flux of 300 L m^−2^ h^−1^ without air scouring. Chemically enhanced backwash (CEB) was performed daily using a 0.05 N NaOH + 50 mg L^−1^ Cl^−^ solution, which was optimized during preliminary testing (Appendix A). The daily CEB cycle was divided into two components: in the first stage, the chemical solution was backwashed through the membrane module for a period of 20 min, which is typical for UF membrane facilities [80]. In the second phase, the backwash feed line was switched from CEB chemicals to RO water and backwashing continued for 20 min to flush remaining chemicals and foulants from the module. Chemical clean–in–place (CIP) was performed at the end of every three–day test using the same protocol as CEB cleaning but with a more aggressive chemical solution of 0.1 N NaOH + 200 mg L^−1^ Cl^−^ solution. Following CIP cleaning, pure water permeability tests were conducted for 1 h using Milli–Q water to assess the permeability restoration. A summary of the operating conditions is presented in Table 1. 

### 2.5. Membrane Performance Measures

#### 2.5.1. Normalized TMP (nTMP) Profiles

Prior to each three–day filtration experiment, pure water permeability testing was conducted with each new membrane module until stability (as determined by TMP variation less than 5%) was reached [81]. For each filtration test, the resulting TMP profiles were normalized against the initial clean water steady–state TMP value. Normalization was performed to reduce the effects of membrane variability when comparing the results following different pretreatment. Slight differences in the porosity, thickness, or pore size during the manufacturing process may result in slightly different clean membrane permeabilities [9]. The nTMP was calculated using Equation (1).
(1)$$\mathrm{nTMP}=\frac{{\mathrm{TMP}}_{t}}{{\mathrm{TMP}}_{o}}$$
where ${\mathrm{TMP}}_{t}$ is the transmembrane pressure at time t (Bar) and ${\mathrm{TMP}}_{o}$ is clean water steady–state transmembrane pressure (Bar). 

#### 2.5.2. Fouling Rates

Hydraulically reversible and irreversible fouling rates (FR) were computed following the approach of Aly et al. [82]. The general form of the FRs is presented below and in the schematic diagram of Appendix A.
(2)$${\mathrm{FR}}_{\mathrm{hr}}=\frac{{\mathrm{TMP}}_{n-1,f}-{\mathrm{TMP}}_{n,i}}{t_{n-1,\;f}-t_{n-1,\;i}}$$
(3)$${\mathrm{FR}}_{\mathrm{hirr}}=\frac{{\mathrm{TMP}}_{n,i}-{\mathrm{TMP}}_{n-1,i}}{t_{n-1,\;f}-t_{n-1,\;i}}$$
where ${\mathrm{FR}}_{\mathrm{hr}}$ is the hydraulically reversible fouling rate (mBar h^−1^), ${\mathrm{FR}}_{\mathrm{hirr}}$ is the hydraulically irreversible fouling rate (mBar h ^−1^), ${\mathrm{TMP}}_{n,i}$ is the measured TMP at the beginning of cycle n (mBar), ${\mathrm{TMP}}_{n-1,f}$ is the measured TMP at the end of cycle n − 1 (mBar), ${\mathrm{TMP}}_{n-1,i}$ is the measured TMP at the beginning of cycle n − 1 (mBar), and $t_{n-1,\;f}$ and $t_{n-1,\;i}$ are the end of cycle filtration time and the beginning of cycle filtration time, respectively. The fouling that is reversible in nature is therefore considered to be that which is recovered by the hydraulic backwash. The fouling that is hydraulically irreversible in nature is then determined as the difference between the initial TMP following hydraulic backwash and the initial TMP of the preceding cycle. The total fouling rate (FR_TOT_) is considered to be the sum of the hydraulically reversible and irreversible fouling rates. The analysis in the current study was based on the three–day average FR_hr_, FR_hirr_, and FR_TOT_ values.

The fouling rates can also be determined for the chemically irreversible fouling, both following CEB and CIP cleanings, as shown in Equations (4) and (5) below and in Appendix A.
(4)$${\mathrm{FR}}_{\mathrm{CEB}}=\frac{{\mathrm{TMP}}_{k}-{\mathrm{TMP}}_{k-1}}{t_{k}-t_{k-1}}$$
where ${\mathrm{FR}}_{\mathrm{CEB}}$ is the irreversible fouling rate following CEB (mBar h^−1^), ${\mathrm{TMP}}_{k}$ is the measured TMP (mBar) following CEB cleaning on day k (for k = 1, 2), ${\mathrm{TMP}}_{k-1}$ is the measured TMP (mBar) following pure water permeability testing on day k − 1 (for k = 1) and following CEB cleaning on day k − 1 (for k = 2), $t_{k}$ and $t_{k-1}$ are the filtration times (measured in h) corresponding to ${\mathrm{TMP}}_{k}$ and ${\mathrm{TMP}}_{k-1}$, respectively. In this study $t_{k}$ was every 22.5 h.
(5)$${\mathrm{FR}}_{\mathrm{CIP}}=\frac{{\mathrm{TMP}}_{3}-{\mathrm{TMP}}_{0}}{t_{3}}$$
where ${\mathrm{FR}}_{\mathrm{CIP}}$ is the irreversible fouling rate following CIP (mBar h^−1^), ${\mathrm{TMP}}_{3}$ is the measured TMP (mBar) following CIP cleaning on day 3, ${\mathrm{TMP}}_{0}$ is the measured TMP (mBar) following pure water permeability testing at the commencement of the test (i.e., day 0), and $t_{3}$ is the length of the filtration test (h). In this study, $t_{3}$ was 67.5 h.

### 2.6. Analytical Methods

Raw and pretreated ORW was analyzed for DOC, pH, zeta potential (ZP), turbidity, UV_254_, and true color. Samples used for DOC and UV_254_ measurement were pre-filtered using a 0.45 µm PES membrane filter (Supor 47 mm, 60043, Pall, Mississauga, ON, Canada). The filter material was selected to minimize the adsorption of organics, as recommended by Karanfil et al. [83]. A total organic carbon (TOC) analyzer (Pheonix 8000, Tekmar–Dohrmann, Cincinnati, OH, USA) was used for analysis of DOC according to Standard Method 5310/5310C, the UV–persulfate oxidation method [84]. Ultraviolet absorbance was measured at a wavelength of 254 nm according to Standard Methods 5910/5910A [84] using a Hach DR6000 UV–Vis spectrophotometer (LPV441.99.00002, Loveland, CO, USA). The feed water SUVA values were determined by dividing the UV_254_ absorbance by the DOC. ZP was measured with a zetasizer nano particle analyzer (Nano ZS Series, Malvern Instruments Ltd., Worcestershire, UK). Turbidity was measured according to Standard Methods 2130/2130B [84] using a Hach 2100AN nephelometric laboratory turbidimeter (4700100, Hach, Loveland, CO, USA). The pH of all samples was measured using a benchtop meter (Symphony B10P, VWR, Mississauga, ON, Canada) and electrode (Red Rod, 89–321–580, VWR, Mississauga, ON, Canada). 

## 3. Results 

### 3.1. Water Quality

The measured water quality indicators for all raw and pretreated waters are presented in Table 2. Raw ORW is characterized by its high true color, SUVA, and relatively high DOC. From the table, it can be seen that all pretreatments were effective in the removal of DOC, color, and UV_254_, although to different extents. CF–DAF resulted in the greatest removal of organics (as measured by DOC) and HPO organics (as measured by SUVA), which is consistent with the findings of earlier studies with this river water [62]. The propensity of CF–DAF pretreatment for removing HPO organics was also reported by Braghetta et al. [56] and Wang et al. [85], who proposed that enhanced HPO interaction occurs at the bubble surface in order to reduce its interface energy. 

Although the mean DOC of the ORW_s_ and ORW_w_ samples were practically identical, the character of the two waters was notably different, especially in terms of hydrophobicity (as quantified in terms of SUVA and UV_254_). The ORW_s_ sample had a UV_254_ absorbance value of 0.225 cm^−1^ yielding a SUVA of 3.41 L mg^−1^ m^−1^, compared to 4.16 L mg^−1^ m^−1^ during the winter season. The true color of the ORW also dropped from 55.3 Pt. Co. units during the winter to 32.3 Pt. Co. units during the summer season. Concerning the CF–IN pretreated waters, several differences could be observed. Regardless of season, pre-coagulation at the 34 mg L^−1^ dose resulted in higher feedwater turbidity compared to coagulation at 27 mg L^−1^. For both seasons, pretreatment at 34 mg L^−1^ resulted in lower residual DOC and UV_254_ levels compared to when 27 mg L^−1^ was used. It can also be observed that, compared to the winter season, CF–IN pretreatment at both dosages during the summer resulted in more HPI feed water (based on a reduced SUVA and UV_254_ absorbance).

### 3.2. Comparison of CF–S, CF–DAF, and CF–IN Pretreatment during Winter Season

The performance of the TMC membrane following pretreatment by CF–DAF, CF–S, and CF–IN is shown in Figure 3a,b, which present the nTMP profiles for all ORW_w_ filtration tests. It should be noted that the CEB was performed daily (after 22.5 h of real filtration time), while the CIP was performed at the termination of each three–day test (after 67.5 h of filtration time). Additionally, while the TMP was measured every 2 s, the data in Figure 3a,b was reduced to intervals of 7.5 min to facilitate interpretation and visualization. Direct filtration of raw ORW resulted in the greatest nTMP increase, reaching a final nTMP of 2.27 after three days. Compared to direct filtration of ORW, CF–DAF and CF–S pretreatment resulted in lower levels of membrane fouling, with final nTMPs of 1.60 and 1.49, respectively. Figure 3b presents a comparison of the CF–IN nTMP profiles with those of raw ORW during the winter season. At the end of the third day, CF–IN pretreatment with 34 mg L^−1^ and 27 mg L^−1^ alum reached final nTMPs of 1.20 and 1.73, respectively. The order of increasing pretreatment effectiveness in terms of final nTMP is therefore: raw ORW < CF–IN (27) < CF–DAF < CF–S < CF–IN (34). It should be noted that, although the nTMP profile for CF–S pretreatment in the current study was slightly lower than that of CF–DAF initially, the final nTMP values at the end of the three–day filtration tests were quite similar.

Selected three–day FRs are presented in Figure 3c,d, while all of the rates are presented in Appendix A. CF–DAF and CF–S pretreatments showed similar levels of total fouling (10.8 ± 0.61 versus 10.2 ± 0.55 mBar h^−1^, respectively) despite CF–DAF pretreatment resulting in a feed water with the lowest residual DOC and SUVA, which is consistent with the results of our previous study [63]. 

A notable difference between the results of the current study and those of past studies using ORW [63] is the performance of CF–IN (34) pretreatment, which resulted in the smallest increase in nTMP compared to all other alternatives (including those with particle separation). It should be noted that the FR_TOT_ for CF–IN (34) pretreatment was not the lowest amongst all alternatives because FR_TOT_ does not incorporate the benefits of the daily CEB cleaning. If instead it had been calculated using the first and last nTMP values, then the order of the fouling rates would have matched that of the final nTMPs. Thus, the final nTMP is expected to be a better indicator of overall fouling performance compared to the FR_TOT_ as calculated in Appendix A. In summary, the most salient point of the tests conducted with ORW_w_ is that the CF–IN (34) condition performed even better than CF–S and CF–DAF pretreatment. 

As a reference, it should be noted that the raw ORW FR_hr_ and FR_hirr_ values were in the same order of magnitude (35.1 ± 1.1 and 1.85 ± 0.5) as those reported by Croft [86] for ORW using a PVDF membrane.

### 3.3. CF–IN Seasonality Assessment

To assess the effects of seasonal water quality differences on CF–IN performance, ORW was sampled in both the summer and winter seasons. The nTMP profiles for CF–IN pretreatment during both seasons at alum dosages of 27 mg L^−1^ and 34 mg L^−1^ are presented in Figure 4a,b, respectively. According to these figures, the relative performance of CF–IN pretreatment at two different doses was dependent on the season. The addition of 27 mg L^−1^ alum (final nTMP of 1.73) resulted in a steeper nTMP increase during the winter season compared to the summer (final nTMP of 1.28), although the greatest difference was only observed on the third day of operation. Furthermore, besides the increased FR_hirr_ observed for 27 mg L^−1^ pretreatment during the winter season (see Figure 5b), the final nTMP was exacerbated by the high FR_CEB_ values, which will be discussed later. At the 34 mg L^−1^ alum dose, the CF–IN pretreatment during the winter season resulted in a lower final nTMP (1.20) compared to the summer season (1.34). The better performance of CF–IN pretreatment at 27 mg L^−1^ compared to 34 mg L^−1^ during the summer season suggests that the coagulant dose for fouling mitigation does not necessarily correspond to that which is better for organics/turbidity removal (See Table 2). This is in agreement with the findings of Kimura et al. [87] and Ding et al. [47].

For a further comparison of the effects of seasonality on CF–IN performance, the hydraulically reversible and irreversible fouling rates for CF–IN (34) and CF–IN (27) pretreated waters during the summer and winter seasons are presented in Figure 5a,b, respectively. CF–IN pretreatment at 34 mg L^−1^ during both the summer and winter season results in hydraulically reversible fouling rates which were statistically the same (overlapping 95% CIs). Although the mean FR_hirr_ is lower for CF–IN (34) during summer season compared to the winter season, the 95% CIs are overlapping, and the observed difference in final nTMP can be attributed to the differences in chemically reversible foulants. It can also be observed from Figure 5a,b that there is a much greater difference in seasonal fouling performance (in terms of nTMP, FR_hr_, and FR_hirr_) for CF–IN pretreatment at 27 mg L^−1^ compared to 34 mg L^−1^. While the mean FR_hr_ and FR_hirr_ values were similar during both the summer and winter season for CF–IN pretreatment at 34 mg L^−1^, pretreatment at 27 mg L^−1^ during the winter season resulted in a significantly lower FR_hr_ and a higher mean FR_hirr_ than during the summer season. In Figure 5c the FR_hirr_ could be quadratically related to the feed water ZP for CF–IN pretreatment, although it must be acknowledged that the inherent variability associated with the FR_hirr_ makes the relationship not so rigidly defined. Maeng et al. [88] similarly reported a quadratic relationship between feed ZP and the hydraulically irreversible fouling resistance, with minimal resistance found at slightly negative ZP values.

A possible explanation for the significantly worsened performance of CF–IN (27W) pretreatment in the current study is the floc characteristics and the nature of their deposition and transport. Based on the ZP measures (Table 2), it appears that for the same coagulation conditions it is more difficult to achieve optimal conditions during the winter season than during the summer. This is especially true for CF–IN (27) pretreatment where the ZP during the winter season was −15.17 mV, while in the summer season reached −8.69 mV. The increased difficulty in achieving optimal conditions during the winter season may be due to the greater feed water turbidity levels (i.e., 4.88 NTU in winter and 1.94 NTU in summer), as Black and Hannah reported that the alum dose required for charge neutralization was significantly affected by the exchange capacity (and hence concentration) of the particle matrix [89]. Additionally, Sharp et al. [90] reported that the higher charge density of HPO organics (which were present in greater quantity during the winter season for ORW according to SUVA measures) necessitates higher coagulant dosages for optimum conditions of charge neutralization. Given the discussion above and the fact that the summer ORW had a lower turbidity and lower HPO NOM content, it is therefore likely that pretreatment at 27 mg L^−1^ is closer to the optimal coagulant dose during the summer than during the winter. 

The higher magnitude ZP that results for CF–IN (27W) is thought to lead to the development of small–sized flocs of low fractal dimension (i.e., loose in nature) [91]. It has been shown by others using the multi-channel h–PES membrane [92,93] that small flocs will deposit more or less homogenously along a greater length of the capillary wall compared to large flocs, which are deposited principally at the capillary dead-end. Therefore, a greater proportion of the fiber capillary is coated by a flocculated suspension which is highly susceptible to compression–this likely explains the rapid TMP increase that occurred following the daily CEB (Figure 4a). Lorenzen et al. [94] also reported that the cake compressibility greatly increased for conditions of higher magnitude ZP. In contrast, the more favorable conditions of ZP reported during the summer season and CF–IN (34) pretreatment during the winter season are thought to have resulted in the formation of larger sized flocs which were deposited at the capillary dead-end. The portion of the fiber affected by flocculated irreversible foulants is thus likely very small, with the remainder of the fiber length experiencing minimal relative flux decline. 

Figure 6 presents the correlation between ZP and the chemically irreversible fouling rates following CEB and CIP cleaning (i.e., FR_CEB_ and FR_CIP_). In Figure 6a, a moderate correlation exists (R^2^ = 0.7623) relating lower magnitude FR_CEB_ to lower magnitude feed water ZP. From Figure 6b, a relatively stronger correlation (R^2^ = 0.8946) is observed demonstrating reduced FR_CIP_ at lower magnitudes of feed water ZP. Electrostatic and morphological differences in floc character are the likely reason behind the observed behavior. Kim [95] investigated the impact of calcium and polymer (pDADMAC) addition to a high SUVA (>5 L mg^−1^ m^−1^) synthetic water ahead of LPM filtration. In this last study, it was also reported that the chemical reversibility of fouling was greatest at low magnitudes of ZP, which were achieved through increased polymer and calcium addition [95]. Kim [95] hypothesized reduced electrostatic repulsion facilitated coiling of the humic NOM molecules, which produced flocs of higher fractal dimension and greater density, which were more amenable to removal by hydraulic and chemical cleaning. 

## 4. Discussion

For ORW_W_ and this membrane type, CF–IN pretreatment at 34 mg L^−1^ was found to be the most suitable pretreatment alternative. This finding suggests that membrane material/type is critical in the selection of an appropriate pretreatment regime, as in our previous study (which used ORW from the same season but a different membrane type) CF–IN pretreatment performed worse than both CF–S and CF–DAF [63]. The importance of membrane material/type is further supported by the fact that for the PVDF membrane used in our previous study the hydraulically irreversible fouling was strongly correlated with the SUVA, while in the current study no such relationship existed (see Appendix A). Therefore, for the same water, it can be concluded that the relative performance of a given pretreatment is uniquely dependent on the membrane material/type. 

For the economic long-term performance of the membrane, control of the hydraulically and chemically irreversible fouling is critical [96]. The pronounced seasonal differences in membrane performance observed in the current study suggest that CF–IN pretreatment will likely benefit from continuous process control/optimization throughout the year. For CF–IN pretreatment of ORW, the FR_hirr_ as well as both the FR_CEB_ and FR_CIP_ seem to be reasonably well correlated with the feed water ZP. These correlations between ZP and measures of irreversible fouling indicate that ZP is likely a critical parameter that may be used for process control. Future research studies should investigate the implementation of online ZP instrumentation for optimized control of CF–IN pretreatment of LPMs. Such real-time monitoring has the potential to lead to significant savings in energy, chemical, and sludge disposal costs [97].

## 5. Conclusions

In this study, an h–PES multi-channel UF membrane was used to directly compare the performance of CF–DAF, CF–S, and CF–IN pretreatment for LPM fouling mitigation in the treatment of a high color, high DOC river water. The main findings for the membrane/membrane configuration used in this study are summarized as follows:For the winter water samples, CF–IN pretreatment at 34 mg L^−1^ could more effectively mitigate membrane fouling compared to CF–DAF and CF–S. This is in contrast to the findings of previous studies using the same highly colored river water but a different membrane type. This finding is an indication of the uniqueness of the interaction between membrane material/configuration and feed water quality and points to the necessity for preliminary testing when verifying the impact of pretreatment for a given membrane;The chemical reversibility of fouling for CF–IN tests seems to be strongly influenced by the ZP of the challenge waters. For CF–IN conditions with lower magnitude ZP, irreversibility of fouling following CEB and CIP was minimized;Distinct seasonal differences in CF–IN performance were observed. For the winter season, CF–IN pretreatment at a high dose led to reduced fouling, while for the summer season the low dose performed better. The better performance at the low dose is linked to the lower turbidity and HPO NOM concentrations in the summer water. The effect of these intra–annual variations should be the subject of future research;CF–S pretreatment of the highly colored ORW resulted in a similar level of fouling as CF–DAF using the h–PES TMC UF membrane. This was despite the fact that CF–DAF exhibited preferential removal of DOC and HPO organics (as measured by SUVA).

## Figures and Tables

**Figure 1 membranes-12-01028-f001:**
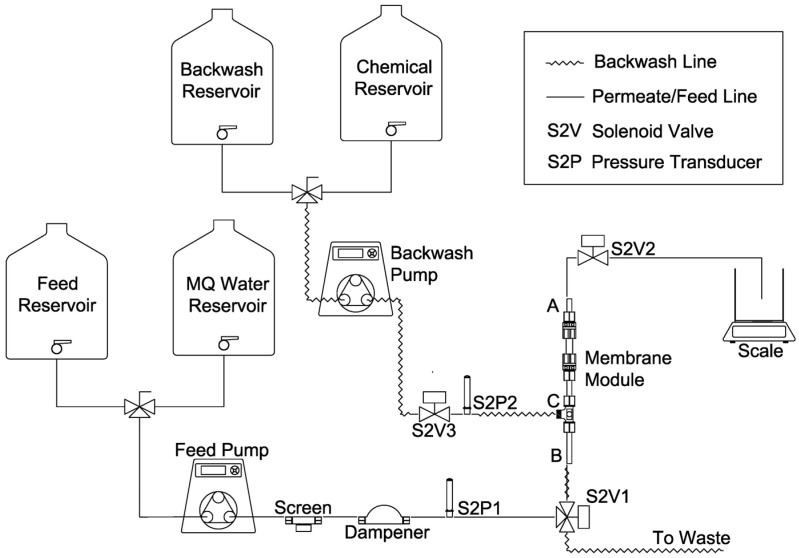
Inside–out filtration apparatus.

**Figure 2 membranes-12-01028-f002:**
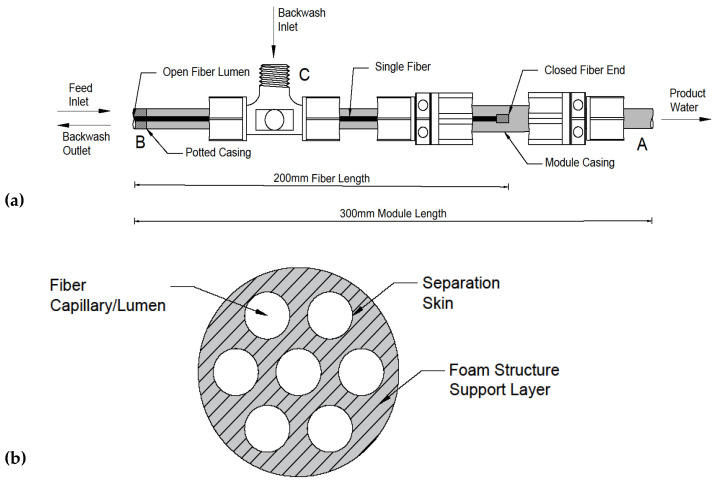
Pressure–driven inside–out membrane: (**a**) module; (**b**) fiber cross–section.

**Figure 3 membranes-12-01028-f003:**
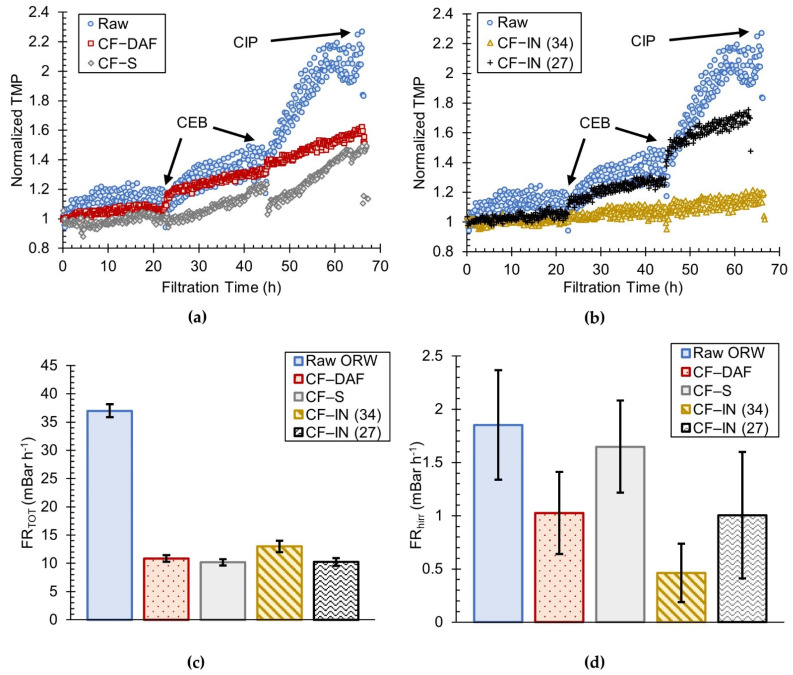
Comparison of ORW_w_ filtration tests: (**a**) raw ORW, CF–S, CF–DAF nTMP profile; (**b**) raw ORW and CF–IN nTMP profile; (**c**) three–day average total FR; (**d**) three–day average hydraulically irreversible FR.

**Figure 4 membranes-12-01028-f004:**
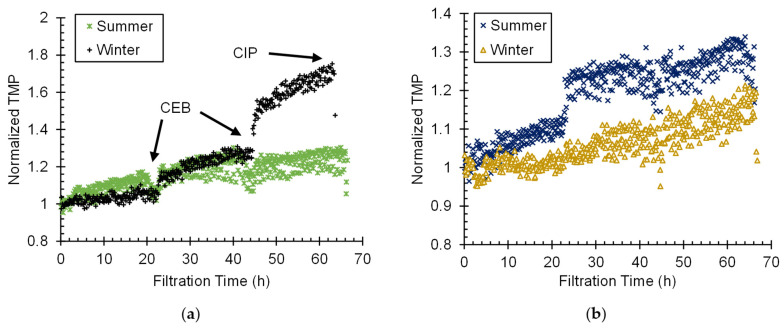
Comparison of summer and winter CF–IN pretreatment at alum dose of: (**a**) 27 mg L^−1^; (**b**) 34 mg L^−1^.

**Figure 5 membranes-12-01028-f005:**
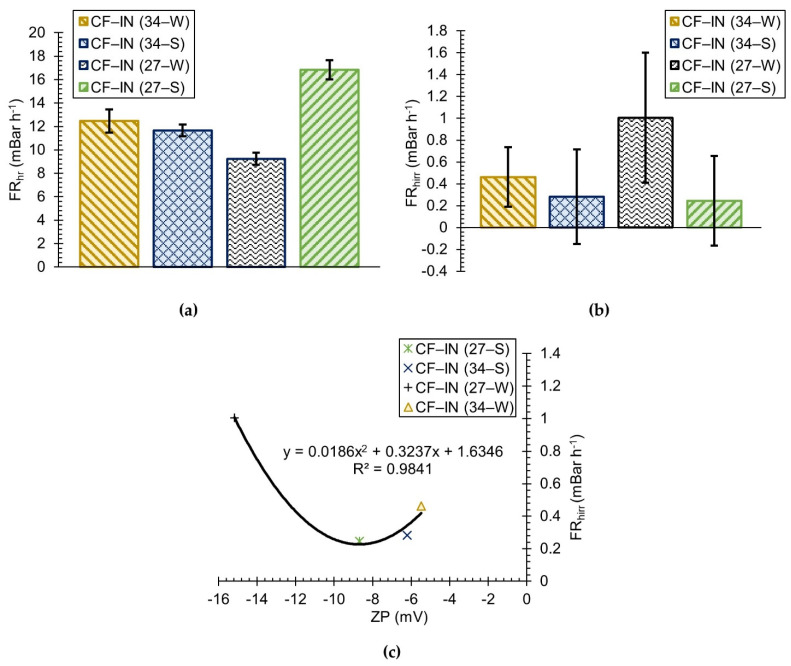
Comparison of seasonal CF–IN fouling rates: (**a**) hydraulically reversible; (**b**) hydraulically irreversible; (**c**) relationship between ZP and hydraulically irreversible.

**Figure 6 membranes-12-01028-f006:**
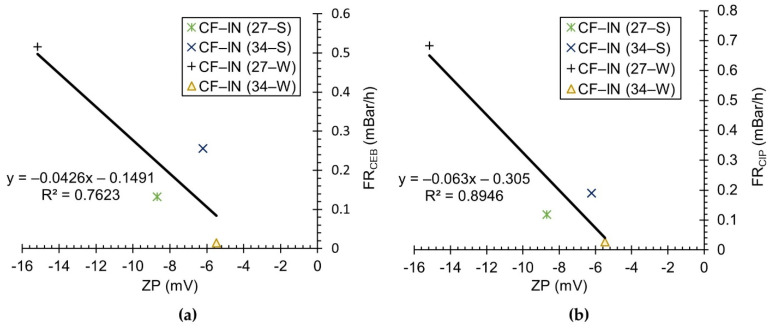
Relationships between zeta potential and chemically irreversible fouling: (**a**) following CEB; (**b**) following CIP.

**Table 1 membranes-12-01028-t001:** Summary of Multi–day Testing Parameters.

Characteristic	Value
Operation	Pressure driven inside–out
Filtration Flux, L m^−2^ h^−1^	80
Backwash Flux, L m^−2^ h^−1^	300
Filtration Cycle Duration, min	30
Backwash Cycle Duration, min	2
CEB Interval, day	1
CEB Chemicals	0.05 N NaOH + 50 mg L^−1^ Cl^−^
CIP Interval, day	3
CIP Chemicals	0.1 N NaOH + 200 mg L^−1^ Cl^−^

**Table 2 membranes-12-01028-t002:** Water Quality Data for Raw and Pretreated ORW.

Parameter	Raw ORW		CF–DAF	CF–S	CF–IN (34 mg L^−1^)		CF–IN (27 mg L^−1^)	
	Winter	Summer	Winter	Winter	Winter	Summer	Winter	Summer
Turbidity (NTU)	4.88 ± 0.01 ^1^	1.94 ± 0.01	0.591 ± 0.02	2.42 ± 0.01	9.11 ± 0.07	5.4 ± 0.01	8.15 ± 0.04	4.76 ± 0.03
pH (unit)	7.52 ± 0.02	7.70 ± 0.03	6.28 ± 0.02	6.08 ± 0.01	6.02 ± 0.01	5.99 ± 0.03	5.98 ± 0.03	6.05 ± 0.03
UV_254_ (cm^−1^)	0.276 ± 0.001	0.225 ± 0.001	0.042 ± 0.001	0.054 ± 0.001	0.051 ± 0.02	0.047 ± 0.01	0.058 ± 0.001	0.055 ± 0.001
True Color (Pt. Co.)	55.3 ± 0.5	32.3 ± 0.5	2 ± 0	2.3 ± 0.5	2 ± 0	2 ± 0	2.3 ± 0.3	3.3 ± 0.3
Zeta Potential (mV)	−22 ± 0.8	−19.3 ± 0.25	−6.31 ± 0.20	−7.23 ± 0.24	−6.21 ± 0.22	−5.48 ± 0.13	−15.17 ± 0.37	−8.69 ± 0.14
DOC (mg L^−1^)	6.64 ± 0.02	6.58 ± 0.01	2.27 ± 0.01	2.67 ± 0.01	2.29 ± 0.01	2.33 ± 0.01	2.55 ± 0.01	2.63 ± 0.01
SUVA (L mg^−1^ m^−1^)	4.16	3.41	1.85	2.03	2.21	1.99	2.26	2.08
DOC Removal (%)	NA	NA	65.81	59.79	65.51	64.59	61.6	60.03
UV_254_ Removal (%)	NA	NA	84.78	80.43	81.52	79.11	78.98	75.56

^1^ µ ± 95% Confidence Interval

## Data Availability

The data presented in this study are available on request from the corresponding author.

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
