# Peer review of "Reduced Low–Pressure Membrane Fouling by Inline Coagulation Pretreatment for a Colored River Water"

_membranes, 2022, doi:10.3390/membranes12111028_

Round 1

Reviewer 1 Report

The manuscript is devoted to study: “The effectiveness of coagulation/flocculation/sedimentation (CF-S), coagulation/flocculation/dissolved air flotation (CF-DAF), and inline coagulation (CF-IN) as technologies for” pretreatment of feed water to control LPM fouling. The correct wording “pretreatment of feed water” but not “LPM pretreatment” (P.1, line 16, P.3, line 62) or “membrane pretreatment” (P.2 line 50 and 70) or “pretreatment of LPMs” (P.2 line 75). Should be corrected.

P.1, lines 31-32: “Low-pressure membranes (LPMs) have become increasingly popular in drinking 31 water treatment (DWT) due to their effectiveness in particle and bacterial separation 32 [1,2].” Refs 1 and 2 are not the best for such a general statement. Even ref.1 is Proceedings of the AMTA/AWWA Membrane Technology Conference. Some recent reviews would be better.

Check the text thoroughly, for example, the following items should be corrected:

11. Arhin, S.G.; Banadda, N.; Komakech, A.J.; Kabenge, I.; Wanyama, J. Membrane Fouling Control in Low Pressure 560 Membranes: A Review on Pretreatment Techniques for Fouling Abatement. Environ. Eng. Res. 2016, 21, 109–120, 561 doi:10.4491/eer.2016.017.

23. Gao, W.; Liang, H.; Ma, J.; Han, M.; Chen, Z. lin; Han, Z. shuang; Li, G. bai Membrane Fouling Control in Ultra-588 filtration Technology for Drinking Water Production: A Review. Desalination 2011, 272, 1–8, 589 doi:10.1016/j.desal.2011.01.051.

Author Response

In response to comment #1, the authors have made the requested wording changes to lines 16, 50, 62, 70, and 75 in order to improve the clarity and consistency for the reader.

In response to comment #2,  in lines 31-32, refs. [1] and [2] were replaced others consisting of several recent (2019-2022) review articles which are better suited for such a general statement.

In response to comment #3, the content of the report was thoroughly reviewed and the noted references citation styles were corrected. 

Reviewer 2 Report

In my opinion the manuscript is well written. The abstract has the necessary information, the introduction has sufficient background, the methodology is adequate, and the results are clearly presented. Also, the conclusions support the results and all references cited are relevant to the research. Therefore, I recommend its publication, but it is important to make a minor change.

 Figures always attract the reader's attention, so I recommend improving the quality of figures 1 and 2 (change the colors of the lines from gray to black).

Author Response

As requested by reviewer #2, the color of the lines within Figures 1 and 2 have been changed from grey to black.